# Better Data for Satellite Super Resolution

**Miguel Castells    Jules Salzinger    Oliver Zendel**
Austrian Institute of Technology Gmbh
Giefinggasse 4, 1210 Vienna, Austria
miguel.castells@student-cs.fr,{jules.salzinger,oliver.zendel}@ait.ac.at

## Abstract

Reliable satellite data is needed for many large-scale tasks in urban planing, agriculture, and disaster relief. However, high resolution satellite data is restricted or expensive. ESA's Sentinel-2 provides free satellite data with global coverage but only at a coarse level of detail. In this work we use super-resolution models trained to create high-resolution versions of Sentinel-2 data. We compare the feasibility of various CLIP embeddings to evaluate similarity between hallucinated satellite data and extend the existing S2-NAIP dataset. We automatically clean unreliable data and add new NIR band data. Our experiments show clear improvement in fidelity and quality of single image cross-sensor super resolution for satellite images.

## 1 Introduction

Earth Observation (EO) is essential for many applications, from urban planning to environmental monitoring. It effectiveness often depends on the availability of **high-quality multispectral satellite imagery**. Among other criteria, a low enough Ground Sampling Distance (GSD, metric size of one pixel) is important for a detailed surface analysis. Access to high-quality satellite and aerial images remains a significant challenge. Some providers offer sub-meter resolution imagery[3, 16], but may suffer from high costs, restricted licensing, coverage limitations, or national security considerations. In contrast, the Sentinel-2[5] project offers global coverage with a revisit time of 5 days under an open-access policy, but has a GSD of $\geq 10$ meters.

**Super-resolution techniques** for satellite imagery promise a way to create training material for reliable ML models based on unreliable upscaled data. Specifically, Single-Image Super-Resolution (SISR) aims to reconstruct a high-resolution (HR) image from a single low-resolution (LR) input.

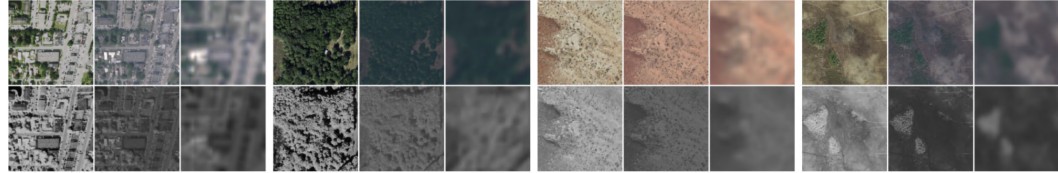

Figure 1: Examples of Sentinel2-NAIP data from our dataset. Top: RGB, Bottom: NIR; Left-to-Right: Original NAIP, Histogram-matched NAIP, Sentinel-2

This task suffers from a distribution gap as HR counterparts for existing LR data cannot be reliably obtained. Large datasets such as S2-NAIP [21] or SEN2NAIP [1] have fueled progress in this domain by providing large amounts of LR/HR pairs. S2-NAIP provides real Sentinel-2 images paired with geographically-aligned NAIP[17] aerial images serving as HR counterparts ("cross-sensor" approach). SEN2NAIP's main split uses NAIP images, which are spectrally aligned and then degraded to mimic Sentinel-2 data ("synthetic" approach).

39th Conference on Neural Information Processing Systems (NeurIPS 2025) Workshop: Reliable ML.

This work improves the typical training and evaluation pipelines in the field of SISR. Notably, we i) compare metrics based on various Contrastive Language-Image Pretraining (CLIP) models; ii) process the S2-NAIP dataset [21] to improve the alignment of its pairs for SISR; iii) add NIR bands to S2-NAIP; and iv) reproduce and assess the recent progress [4, 6] proposing synthetic pretraining with cross-sensor fine-tuning as an efficient paradigm for SISR.

## 2  Literature Review

LR/HR pairs for SISR can be constructed by artificially degrading HR images into corresponding LR inputs (synthetic datasets) or by aligning low- and high-resolution images acquired from different sensors such as in [21] (cross-sensor datasets). Recent efforts for synthetic datasets [8, 23, 27] proposed sophisticated degradation methods to simulate cross-sensor effects (e.g., based on U-Net [1]). This allows for better LR/HR alignment, but distribution gaps still remain. True cross-sensor data ensures that the model learns from the correct input distribution, but introduces new challenges: **geometric and radiometric discrepancies** [9], and visible **temporal misalignment** (cloud coverage, disaster damage, seasonal variations...) due to LR/HR data from different capture dates. Both dataset types require careful preprocessing and alignment strategies to handle these unreliable uncertainties.

Another challenge of SISR is the difficulty of designing consistent metrics for benchmarking. Although Peak Signal-to-Noise Ratio (PSNR) and Structural Similarity Index Measure (SSIM) are widely adopted, they often fail to reliably capture perceptual quality [9, 21], which is more robust to misalignments. Recent studies have proposed alternative metrics, including CLIP-based methods [21, 11, 10], the OPENSR metrics [2], and no-reference image quality assessment approaches [9], which aim to provide more robust and perceptually aligned evaluations.

## 3  CLIP comparison

To address the inconsistencies of common image quality metrics such as PSNR and SSIM, Wolters et al. [21] proposes a novel metric named **CLIPScore**. It computes the cosine similarity between the semantically rich CLIP embeddings of the two images. This promises better metrics for complex scenes such as aerial imagery. However, CLIPScore exhibits high scores with low variance across various classes of aerial images (ranging from 0.7 to 0.99), and surprisingly high scores between highly dissimilar images (e.g., a score of 0.67 for dog vs. computer).

We assess the performance of new CLIP alternatives trained for remote sensing tasks [15, 10, 11] using two datasets: **AID** [22]; a dataset of aerial images and **CelebA** [12], a dataset of face images. We use 100 random images from three of AID's scene types: *School*, *Industrial*, and *Desert*. These classes are chosen based on their perceptual similarity: *School* and *Industrial* are expected to be more similar than *Desert* scenes. An additional 100 random CelebA images provide samples of a clearly distinct image domain. Using different CLIP models, we compare the average CLIPScore between images from the School class (intra-class similarity) and every image across the other selected classes (inter-class similarity). The resulting score histograms allow us to evaluate each model's behaviour.

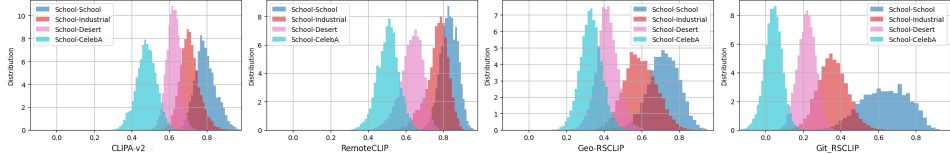

Figure 2: Distribution of CLIPScore across different image subsets from AID and CelebA

Our results can be seen in **Figure 2**. Unlike CLIPA-v2 and Git-RSCLIP, some models show a significant overlap in the distribution of CLIPScore values between face images and landscape images, indicating less distinct separation between these semantically different classes. Moreover, RemoteCLIP, CLIPA-v2, and Geo-RSCLIP exhibit a large overlap in their score distributions over various remote sensing classes. This suggests that these models may struggle to distinguish between different landscape categories, potentially limiting their effectiveness in tasks that require fine-grained differentiation. For SISR, the images to compare are very similar semantically, requiring a high capability to discriminate between different classes and within a given class.

In contrast, Git-RSCLIP demonstrates better class separation, and a greater variance in CLIPScore within the same class. This higher intra-class variance implies that it is better at distinguishing subtle differences within similar images making it particularly well-suited for SISR.

## 4 Dataset

We process S2-NAIP [21] to make it a cross-sensor dataset suitable for SISR, with RGB and NIR bands. The original dataset consists of 1.2 million $128 \times 128$ HR NAIP images ($\approx 2.4$ m GSD) with corresponding $32 \times 32$ LR Sentinel-2 time-series images ($\approx 9.5$ m GSD). Those time-series varies from 32 to hundreds of observations which may contain clouds, seasonality changes such as snow cover, and low reflectance images. Prior work uses random images from the time series, thus introducing noise in the training process. We improve this by selecting the perceptually closest Sentinel-2 image from the time-series (for each NAIP image).

**Pre-processing** As a first step, we clean up the dataset by discarding pairs where NAIP images contain black pixels along the border, as well as NAIP images with no associated Sentinel-2 time series ($\approx$1 million remaining pairs). In each Sentinel-2 time series, we similarly discard entries with black borders. We also remove most of the cloudy and low-reflectance observations by keeping in the time series only the 16 images with the highest PSNR with respect to the HR reference (after resampling). Though imperfect, this step considerably reduces the number of cloudy images in the dataset. We then download and reproject the NIR bands for the NAIP dataset (the Sentinel-2 data in S2-NAIP already contains all bands) using their unique tile identifiers from the dataset.

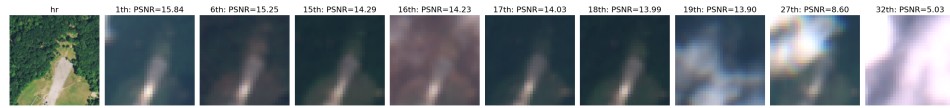

Figure 3: The left-most image is the high-resolution (HR) reference. Adjacent images depict kept low-resolution (LR) images from the time series, chosen according to their PSNR scores compared to the HR image. Lower PSNR images often correspond to cloudy observations.

**Selecting the Closest LR Image** The main differences between HR and LR images are due to the domain gap and temporal discrepancies. The domain gap refers to the shift in pixel value distributions due to differences in sensor characteristics and surface reflectance [9]. Temporal discrepancies arise from landscape changes such as snow cover, seasonal variations, or harvest cycles.

To capture the similarity between LR and HR images, we compute two types of distances:

- The raw distance between each LR image and its corresponding HR image, which captures overall differences including both the domain gap and perceptual discrepancies.
- The distance between each LR image and the HR image after histogram matching, using scikit-image[18] and the LR image as the reference. This metric emphasizes temporal discrepancies while reducing the domain gap, but cannot be used alone as it will favour very low- or high-reflectance images remaining in the data.

For each pair $(HR, [LR]_n)$, and for each $j \in [\![1, n]\!]$, we compute the combined score:

$$\text{Score}_j = \alpha \cdot \text{d}(HR, LR_j) + (1 - \alpha) \cdot \text{d}(HR_{\text{histm}_j}, LR_j)$$

where $\alpha = 0.7$ is chosen empirically and $HR_{\text{histm}_j}$ denotes the high-resolution image histogram matched to the $j$-th low-resolution image as reference.

After selecting the best LR image, we form image pairs for SISR training by applying histogram matching to the HR image using the chosen LR as reference thus reducing radiometric discrepancies (see Figure 1).

**Manually improving the test set** Training and evaluating models on our new dataset validation set reveals a high variance in the results. Approximately 200 of the 7571 images from our test set yield outlier PSNR values below 20 dB. These are failure cases from our selection process (e.g. clouds

or seasonality differences). We fix this by manual selection leading to 166 refined and 27 removed entries (due to the lack of any suitable low-resolution pair).

# 5 Impact of Pretraining

We assess our dataset by training an ESRGAN [19] as our reference model. The training procedure is conducted in two steps following recent trends [4, 6]: we first pre-train the model using the SEN2NAIPv2 dataset [1], then we fine-tune the model on our dataset.

The SEN2NAIPv2 dataset exists in two versions for two different methods of synthetic alignment (histogram matching and U-Net based). We evaluate the impact of pretraining on model performance under three conditions: (1) pretraining on the SEN2NAIPv2-HM variant (using Histogram Matching), (2) pretraining on SEN2NAIPv2-UNet, and (3) training from scratch without any pretraining. This comparison allows us to assess how different synthetic data generation strategies affect performance.

Table 1: Quantitative results on our test set. Best scores are highlighted in **bold**.

| Model | PSNR-RGBN (dB) ↑ | SSIM-RGBN ↑ | PSNR-RGB (dB) ↑ | SSIM-RGB ↑ | Git-RSCLIP-RGB ↑ | LPIPS-RGB ↓ |
|---|---|---|---|---|---|---|
| *No Pretraining* **+ Finetuning** | 25.885 | 0.548 | 27.860 | 0.594 | 0.670 | 0.358 |
| Pretraining SEN2NAIPv2-HM | 25.718 | 0.593 | 27.695 | 0.650 | 0.496 | 0.480 |
| Pretraining SEN2NAIPv2-HM **+ Finetuning** | **27.514** | 0.588 | **28.308** | 0.603 | 0.734 | 0.343 |
| Pretraining SEN2NAIPv2-UNet | 27.112 | **0.630** | 27.912 | **0.654** | 0.492 | 0.470 |
| Pretraining SEN2NAIPv2-UNet **+ Finetuning** | 27.432 | 0.587 | 28.249 | 0.601 | **0.743** | **0.340** |

**Table 1** presents the results of the models after 20k pretraining iterations and 8k fine-tuning iterations.

Pretraining shows a positive impact on all metrics except SSIM, highlighting the inconsistency of this metrics. The performance difference between `HM` and `UNet` versions are relatively small. Figure 4 shows HR outputs from each model for the same LR input. It is clear that finetuning using our data vastly improves details and pre-training is necessary to reduce generation artifacts. Git-RSCLIP in **Table 1** succeeds to properly reflect these quality changes while PSNR and SSIM fail.

| LR | No Pr.+FT | Pr. HM | Pr. HM+FT | Pr. Unet | Pr. Unet+FT | GT |
|---|---|---|---|---|---|---|

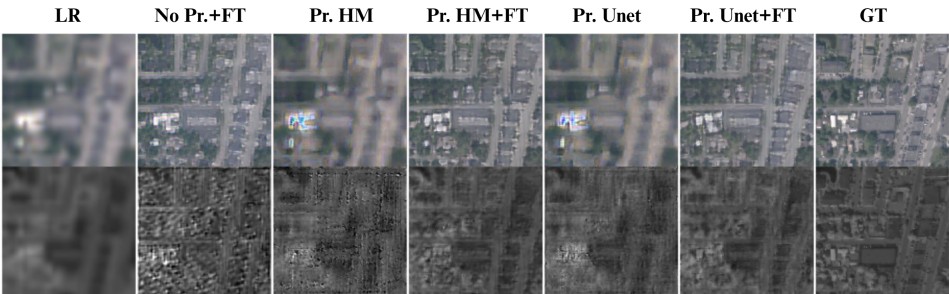

Figure 4: Outputs of super resolution models, same order as Table 1. **LR** stands for low-resolution, **Pr.** stands for Pretraining, **FT** stands for Fine-tuning, **HM** stands for Histogram-Matching Result, and **GT** stands for Ground-Truth.

# 6 Discussion

**Limitations**    One limitation of this study lies in the cloud removal process, which could be improved by leveraging additional spectral bands or more recent cloud detection techniques. Additionally, the current use of histogram matching for aligning image characteristics could be improved (e.g., using the U-Net employed in SEN2NAIPv2 [1]).

**Future Directions**    Future work could focus on extending the dataset to a global scale to improve the model's generalisation capabilities. Incorporating zero-shot classification approaches—such as those based on CLIP—to estimate the distribution of landscape classes could help balance or stratify training data according to real-world land cover distributions. The current work rewards super-resolving homogeneous landscapes like oceans similarly to highly heterogeneous areas like urban environments, which exhibit richer high-frequency details.

## Acknowledgments

This research has been supported by the EMERALD project (Enhanced Multi-resolution Earth-observation using Robust and Advanced Learning for Environmental Dynamics), funded by the Austrian Research Promotion Agency (FFG).

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

# Appendix

## 7 Online resources

Please see our companion repo at `https://github.com/ozendelait/better_s2_naip` for scripts to automatically create the NAIP-NIR extension of S2-NAIP, the semi-automatic selection process for best single image correspondence, and the manual correction data for S2-NAIP.

### 7.1 CLIP Embedding Variants

Our paper compares multiple CLIP embeddings which can be used to represent satellite data in a high-dimensional feature space.

#### 7.1.1 CLIPA-v2

The goal of CLIPA-v2 is to provide a framework that can train CLIP efficiently and effectively at scale. It is trained on *Datacomp*[7] , a large benchmark designed for multimodal dataset curation, consisting of 12.8 billion image-text pairs from Common Crawl.

#### 7.1.2 RemoteCLIP

RemoteCLIP is a vision-language foundation model for remote sensing that aims to learn robust visual features with rich semantics and aligned text embeddings for seamless downstream applications. It is trained on a comprehensive dataset that combines a wide range of remote sensing datasets (RSITMD[25], RSICD[13], UCM[24]) addressing the scarcity of pre-training data in remote sensing.

#### 7.1.3 Geo-RSCLIP

The goal of Geo-RSCLIP is to perform zero-shot classification tasks, achieving significant improvement in Remote Sensing Cross-Modal Text–Image Retrieval and in Semantic Localization tasks. It is trained on *RS5M*[28], a large-scale remote sensing image-text paired dataset constructed by filtering large-scale image-text datasets and captioning RS datasets with a pretrained model.

#### 7.1.4 Git-RSCLIP

The goal of Git-RSCLIP is to explore the application of the Git-10M dataset to pretrain a vision-language foundation model using a contrastive learning framework. It is trained on *Git-10M*[10], the largest-scale remote sensing image-text dataset with extensive geographical diversity and metadata, overcoming the limitations of previous small-scale datasets and providing a robust foundation for training generative models.

### 7.2 Evaluation Metrics

These evaluation metrics are used in our work to measure visual similarity between satellite data.

#### 7.2.1 Peak Signal-to-Noise Ratio (PSNR)

The Peak Signal-to-Noise Ratio (PSNR)[14] is a widely used metric to evaluate the quality of a reconstructed or compressed image compared to a reference image. It is defined using the Mean Squared Error (MSE) between the reference image $I$ and the distorted image $\hat{I}$ as

$$\text{MSE} = \frac{1}{mn} \sum_{i=0}^{m-1} \sum_{j=0}^{n-1} \left( I(i,j) - \hat{I}(i,j) \right)^2, \tag{1}$$

where $m \times n$ is the image size. The PSNR in decibels (dB) is then given by

$$\text{PSNR} = 10 \cdot \log_{10} \left( \frac{MAX_I^2}{\text{MSE}} \right), \tag{2}$$

with $MAX_I$ denoting the maximum possible pixel intensity (e.g., 255 for 8-bit images).

### 7.2.2 Structural Similarity Index (SSIM)

The Structural Similarity Index (SSIM)[20] is a perceptual metric that measures the similarity between a reference image $I$ and a distorted image $\hat{I}$ by considering structural information. It combines luminance ($l$), contrast ($c$), and structure ($s$) comparisons into a single measure:

$$\text{SSIM}(I, \hat{I}) = \frac{(2\mu_I\mu_{\hat{I}} + C_1)(2\sigma_{I\hat{I}} + C_2)}{(\mu_I^2 + \mu_{\hat{I}}^2 + C_1)(\sigma_I^2 + \sigma_{\hat{I}}^2 + C_2)}, \tag{3}$$

where $\mu_I, \mu_{\hat{I}}$ are mean intensities, $\sigma_I^2, \sigma_{\hat{I}}^2$ are variances, and $\sigma_{I\hat{I}}$ is the covariance. The constants $C_1$ and $C_2$ stabilize the division, and SSIM values range from $-1$ to $1$, with $1$ indicating perfect structural similarity.

### 7.2.3 Learned Perceptual Image Patch Similarity (LPIPS)

The Learned Perceptual Image Patch Similarity (LPIPS)[26] is a deep-learning-based metric that evaluates the perceptual similarity between a reference image $I$ and a distorted image $\hat{I}$. It computes distances in feature space using a pretrained convolutional neural network (e.g., AlexNet, VGG) as

$$\text{LPIPS}(I, \hat{I}) = \sum_l \frac{1}{H_l W_l} \sum_{h,w} \|w_l \odot (f_l^I(h,w) - f_l^{\hat{I}}(h,w))\|_2^2, \tag{4}$$

where $f_l^I$ and $f_l^{\hat{I}}$ are normalized feature maps at layer $l$, $w_l$ are learned channel-wise weights, and $H_l, W_l$ denote spatial dimensions. Lower LPIPS values indicate higher perceptual similarity, making it well aligned with human visual judgments compared to traditional metrics.

### 7.3 Histogram Matching

Histogram matching is a technique used to adjust the pixel intensity distribution of a source image $I$ so that it matches the histogram of a reference image $R$. Formally, it relies on cumulative distribution functions (CDFs) of the two images:

$$\hat{I}(x) = F_R^{-1}(F_I(x)), \tag{5}$$

where $F_I$ and $F_R$ denote the CDFs of the source and reference images, respectively, and $F_R^{-1}$ is the inverse CDF. This process ensures that the transformed image $\hat{I}$ has an intensity distribution statistically similar to $R$, improving visual consistency across images.

For this paper, we use the consistent library *scikit-learn v0.25.2*[18]

## 7.4 More results from our pipeline

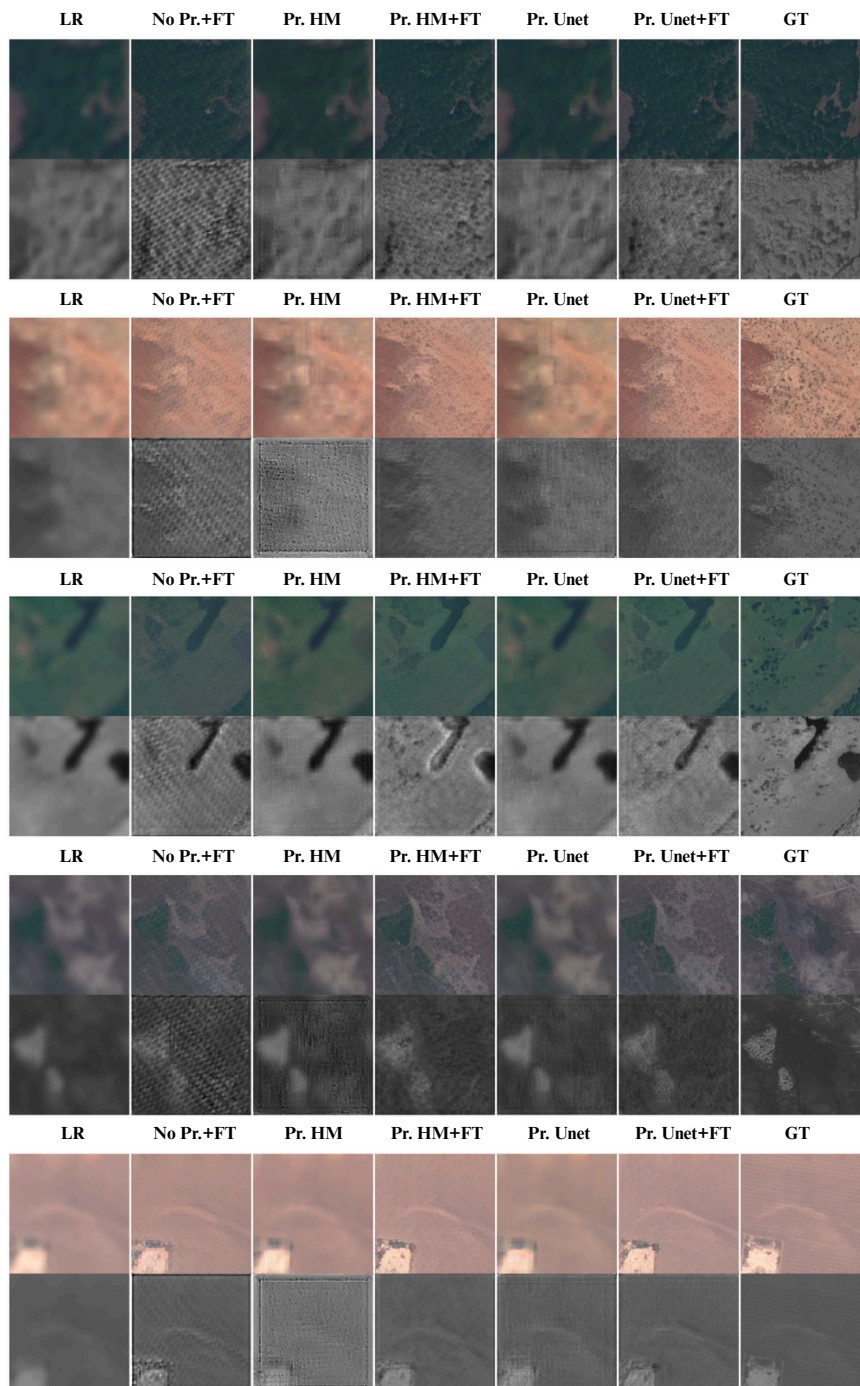

Figure 5: More outputs of super resolution models, same order as Table 1. **LR** stands for low-resolution, **Pr.** stands for Pretraining, **FT** stands for Fine-tuning, and **HM** stands for Histogram-Matching Result

