# OpenReview forum: "Better Data for Satellite Super Resolution"
_NeurIPS.cc/2025/Workshop/Reliable_ML — NeurIPS 2025 - Reliable ML Workshop_

### Official Review · Reviewer_zbGC · 2025-09-10
**On Single-Image Super-Resolution (SISR)**

**Rating:** 6
**Confidence:** 3

**Review:**

1. The work aims to add High Resolution (HR) images to the Sentinel-2 (S2) dataset, which are Low Resolution (LR) images, for downstream tasks. They obtain the HR images from NAIP dataset and the issue is on how to match images between the LR and HR datasets, since they might be of the same location but under different environmental conditions. The paper suggests a new metric for benchmarking and obtain matching between NAIP and S2 dataset for high-quality HR-LR pairs in training SISR.

2. Strengths: The work possibly contributes a high quality HR-LR image pair dataset, subject to being publicly released.

3. Weaknesses: The weaknesses of other metrics are not explained. The strengths of the new metric are also not clear. The score used in line 102 is quite heuristic and not motivated well. Other than the contribution of a new S2-NAIP dataset, the contributions are not clear or convincing.

4. Suggestions: explain PSNR, SSIM, histogram matching in the paper (or appendix). CLIP models and their variations should also be explained.

---

### Official Review · Reviewer_tuKa · 2025-09-20
**Methodological contribution to satellite image applications**

**Rating:** 7
**Confidence:** 3

**Review:**

This paper addresses the challenge of obtaining high quality satellite data at low cost. The freely available data are coarse, and high resolution data are expensive or unavailable. The authors compare CLIP embeddings used to evaluate satellite image cleaning, demonstrate a preprocessing method, and conduct experimentation on their resolution-enhancing procedure.

The work addresses a well-motivated challenge in the remote sensing community. The CLIP model evaluation is also thorough and insightful. The primary contribution seems to be this evaluation and comparison of existing methods, where they show that GitRSCLIP offers the best performance and variance. Their data processing pipeline is also a nice contribution, in particular the proposed method for temporal alignment. The manual selection of test set images raises concerns of selection bias and validity that need to be addressed. The work also references unreliable data several times throughout; spelling this out explicitly would be valuable. In particular, the specific goal of the analysis was not clear to me.